There are amendments to this paper

# G9a regulates breast cancer growth by modulating iron homeostasis through the repression of ferroxidase hephaestin

Ya-fang Wang[1,2], Jie Zhang[3], Yi Su[1], Yan-yan Shen[1], Dong-xian Jiang[4], Ying-yong Hou[4], Mei-yu Geng[1], Jian Ding[1] & Yi Chen[1]

G9a, a H3K9 methyltransferase, shows elevated expression in many types of human cancers, particularly breast cancer. However, the tumorigenic mechanism of G9a is still far from clear. Here we report that G9a exerts its oncogenic function in breast cancer by repressing hephaestin and destruction cellular iron homeostasis. In the case of pharmacological inhibition or short hairpin RNA interference-mediated suppression of G9a, the expression and activity of hephaestin increases, leading to the observed decrease of intracellular labile iron content and the disturbance of breast cancer cell growth in vitro and in vivo. We also provide evidence that G9a interacts with HDAC1 and YY1 to form a multi-molecular complex that contributes to hephaestin silencing. Furthermore, high G9a expression and low hephaestin expression correlate with poor survival of breast cancer are investigated. All these suggest a G9a-dependent epigenetic program in the control of iron homeostasis and tumor growth in breast cancer.

[1] Division of Anti-Tumor Pharmacology, State Key Laboratory of Drug Research, Shanghai Institute of Materia Medica, Chinese Academy of Sciences, Shanghai 201203, China. [2] University of Chinese Academy of Sciences, No. 19A Yuquan Road, Beijing 100049, China. [3] Key Laboratory of Systems Biology, CAS Center for Excellence in Molecular Cell Science, Institute of Biochemistry and Cell Biology, Shanghai Institutes for Biological Sciences, Chinese Academy of Sciences, Shanghai 200031, China. [4] Department of Pathology, Zhongshan Hospital, Fudan University, Shanghai 200032, China. Correspondence and requests for materials should be addressed to M.-y.G.(email: mygeng@simm.ac.cn) or to J.D.(email: jding@simm.ac.cn) or to Y.C.(email: ychen@simm.ac.cn)

As a histone methyltransferase (HMTase), G9a contains a SET (Su (var), Enhancer of Zeste, Trithorax) domain and localizes in euchromatin regions where it mediates the methylation of histones H3K9 and H3K27[1,2]. In particular, H3K9 methylation by G9a is an integral component of transcriptional repression for many genes during diverse biological processes. G9a is essential for early mouse embryo development and embryonic stem cell differentiation[2]. Moreover, a large body of evidence indicates a role for G9a in tumorigenesis. G9a is highly expressed in many cancers, including human bladder, lung, colon and claudin-low breast cancer, compared with its expression in normal tissue[3–5]. Its repressive role in E-cadherin expression makes it a marker of aggressive ovarian cancer and endometrial cancer. The deregulated function of G9a in cancers suggests that it may be a viable therapeutic target[6]. However, the tumorigenic role of G9a in breast cancer is still far from clear.

Cellular iron homeostasis is not only critical for biological processes in normal cells, but also contributes to both the initiation and growth of tumors. Iron deficiency can cause growth arrest and cell death, whereas excessive iron generates free radicals that damage DNA, lipid membranes and proteins[7,8]. Recent work has also shown that iron plays a role in the tumor microenvironment and metastasis. The pathways of iron acquisition, efflux, storage and regulation are all perturbed in cancer, suggesting that the reprogramming of iron metabolism is a central aspect of tumor cell survival[9–11]. Therefore, molecules that regulate iron metabolism are potential therapeutic targets. Hephaestin (HEPH) is a ceruloplasmin (CP) homologue that plays a critical role in intestinal iron absorption. It converts iron in reduction state II ($Fe^{2+}$) into oxidation state III ($Fe^{3+}$) and mediates iron efflux in concert with the ferric exporter ferroportin (FPN) to transport iron across the basolateral membrane[12,13]. HEPH has been detected in colon, spleen, kidney, breast, placenta and bone trabecular cells[14–16], but its role has yet to be established. It remains unclear whether HEPH concentration has any impact on iron in breast tissue and breast cancer growth.

In the present study, we discover that G9a represses HEPH expression, changes cellular iron homeostasis, and stimulates breast cancer growth. We show that the regulation of iron metabolism contributes to the tumorigenic activity of G9a, suggesting the novel function of G9a in controlling cellular iron metabolism and tumor growth. We also endeavor to elucidate the mechanisms underlying the HTMase G9a in HEPH transcriptional repression.

## Results

**G9a plays an important role in breast cancer proliferation.** We initially investigated the effect of G9a expression on breast cancer growth. Specific short hairpin RNAs (shRNAs) or small interfering RNAs (siRNAs) were used to knockdown G9a expression in MCF-7, MDA-MB-231, S1, SK-BR-3 and MDA-MB-435 cell lines. Compared with the parental cells, the cells that stably suppressed G9a expression grew more slowly and possessed a reduced capacity for colony formation (Fig. 1a). In contrast, overexpressed G9a promoted breast cancer cell proliferation in vitro (Fig. 1b). To further substantiate these observations the G9a-specific inhibitors UNC0638 and BIX-01294 were used. These inhibitors also significantly suppressed breast cancer cell proliferation, with the $IC_{50}$ values as several micromoles (Fig. 1e). Furthermore, the breast cancer cells were arrested in G1 phase when G9a was suppressed by shRNA or G9a inhibitors (Fig. 1c). Western blotting analysis showed that G9a inhibition led to a marked down-regulation of cyclin D1, c-Myc and E2F1, and an upregulation of p21, which are collectively required for cell cycle progression from G1 phase to S phase

(Fig. 1d). We also employed a xenograft mouse model to query whether G9a expression is required for tumour growth in vivo. S1 cells with different levels of G9a were subcutaneously inoculated into nude mice, and all the mice developed palpable tumors within 7 days; however, silencing G9a impaired S1 tumor growth (Fig. 1f and Supplementary Fig. 1b). Therefore, we think that G9a is essential for the promotion of breast cancer growth.

**G9a represses HEPH expression in breast cancer.** Given the role of G9a in the epigenetic control of transcription, we performed microarray profiling to identify potential G9a target genes involved in breast cancer cell proliferation. The data revealed that ferroxidase *HEPH* is among the most significantly upregulated transcripts by G9a inhibition (Fig. 2a), for which no function in breast cancer has been ascribed so far. We substantiated this result by detecting the mRNA and protein levels of HEPH in G9a-silenced cells. As with the microarray profiling data, HEPH was noticeably up-regulated in G9a-knockdown breast cancer cells (MCF-7, MDA-MB-231, ZR-75-30, S1, SK-BR-3 and MDA-MB-435) compared with the control (Fig. 2b and Supplementary Fig. 1a, 6a, 9). In contrast, overexpression of G9a reduced the mRNA and protein levels of HEPH in breast cancer cells (Supplementary Fig. 1c, 6b, 9). The G9a-specific inhibitors UNC0638 and BIX-01294 also increased HEPH expression in a dose- and time-dependent manner accompanied by decreasing H3K9-me2 in the cells (Fig. 2c and Supplementary Fig. 1d, 6c, 9).

Moreover, HEPH levels were determined in human normal mammary epithelial cell MCF10A and 20 breast cancer cell lines in which G9a were detected (Fig. 2d). The results showed a noteworthy inverse correlation between G9a and HEPH expression, independent of breast tumor type (Fig. 2e). To find out whether G9a regulates HEPH expression in vivo, we examined HEPH expression in tumor tissues from the G9a shcon and depletion xenografts. Consistent with our in vitro results, we found that G9a depletion xenografts had higher levels of HEPH protein in the tumor tissue (Supplementary Fig. 2a) compared with the shcon tumors.

We next attempted to determine whether G9a expression inversely correlates with HEPH levels in human breast cancer patients. The representative immunohistochemistry analysis of 75 breast cancer specimens revealed inverse staining patterns between G9a and HEPH expression in breast cancer tissues, independent of tumor type (tested by Pearson's nonparametric correlation test, correlation coefficient: −0.678, $P < 0.05$; Fig. 2f). All these data strongly suggest that G9a inhibits HEPH expression in breast cancer.

**Depletion of G9a increases HEPH expression and activity.** HEPH is an integral membrane protein with a single membrane-spanning domain at its C-terminus. It can directly contact with a membrane-bound iron exporter FPN that transports ferric iron through the membrane[16,17]. However, recent immunocytochemical studies have shown that the protein is located at intracellular and supranuclear sites, rather than on the plasma membrane[17]. Cytoplasm localization of HEPH has confirmed its involvement in the intracellular oxidation of iron[18]. Therefore, we examined the detailed localization and function of HEPH by immunocytochemistry (ICC) analyses, cell component separation and ferroxidase activity assays. The ICC results showed that HEPH really does exist on the membrane and in the cytoplasm, which is consistent with previous studies[19]. Moreover, according to our analysis, the positive fluorescence of HEPH greatly increased both on the membranes and in the intracellular sites in G9a-knockdown or pharmacologically inhibited breast cancer cells

(Fig. 3a). Furthermore, western blotting analysis revealed that the increased levels of HEPH in G9a-knockdown cells and enzyme-inhibited cells were most apparent on the cell membrane, whereas the cytoplasmic concentrates were relatively low (Fig. 3b).

As HEPH protein works in concert with FPN to facilitate iron transport across the basolateral membrane through its

ferroxidase activity, we next determined whether HEPH protein activity in cells is enhanced by G9a inhibition. We measured HEPH activity using *p*-phenylenediamine (pPD) oxidation and the ferrozine assay[20]. As shown in Fig. 3c, the in-gel HEPH pPD oxidase activity, confirmed by densitometric measurements of the pPD signal and which measured

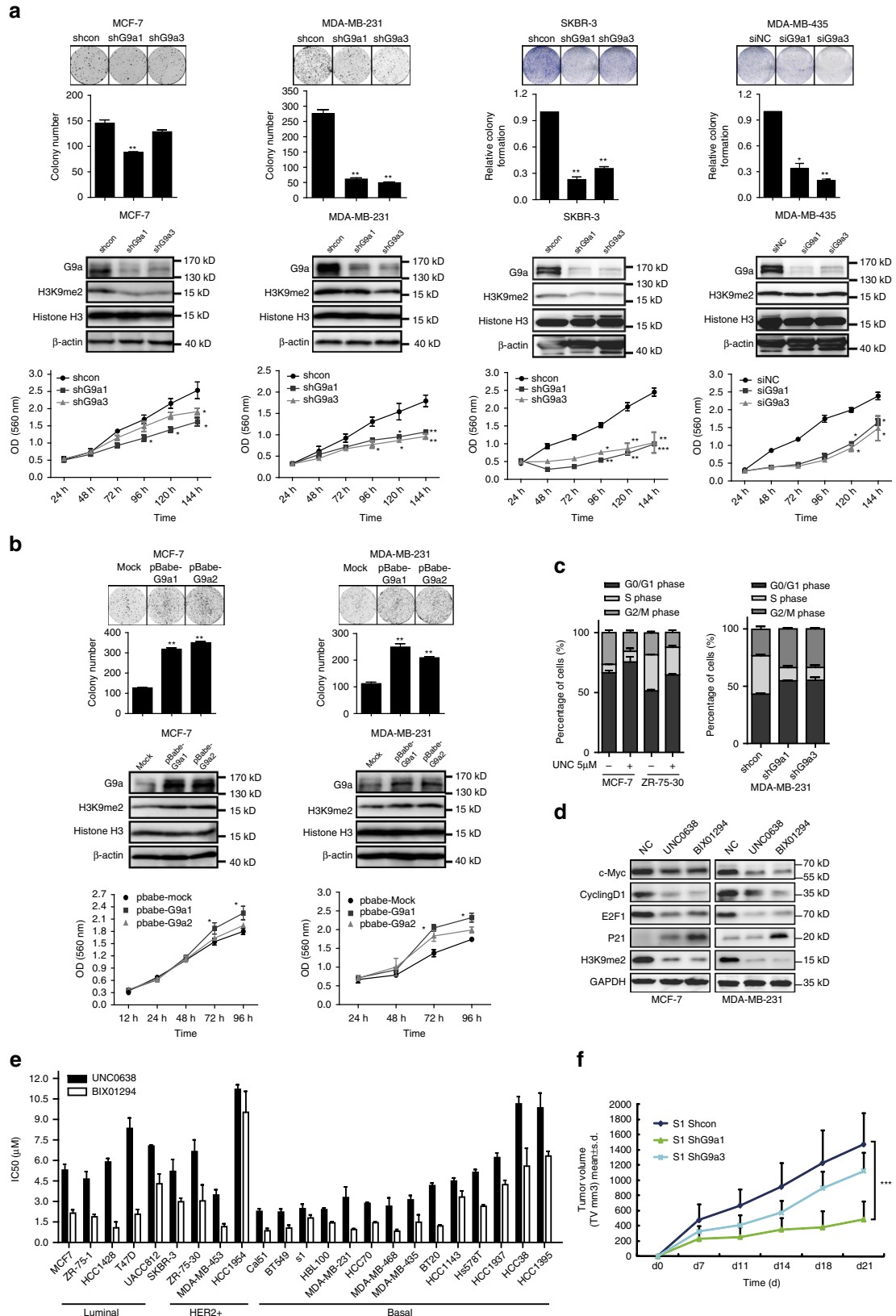

by the ferrozine assays, almost doubled in G9a knockdown MCF-7 and MDA-MB-231 cells compared with the control. Conversely, pPD oxidase activity was reduced in G9a-overexpressed cell lines accompanying with HEPH reducing (Supplementary Fig. 2c). Moreover, HEPH ferroxidase activity which was measured by the in-gel and in-tube ferroxidase activity assays carried out in the G9a-knockdown or overexpressed cell extracts yielded similar results (Fig. 3c and Supplementary Fig. 2c).

HEPH facilitates the oxidation of ferrous iron ($Fe^{2+}$) and decreases the content of intracellular metabolic iron. Owing to its pivotal role as a regulator of iron metabolism, we investigated the effect of increased HEPH on G9a-depleted cells and determined whether HEPH was functional in the labile iron pool (LIP) and in ferritin H chain expression[21]. We initially investigated the LIP by calcein-AM (calcein-acetoxymethyl) assay. As expected, in G9a-depleted and inhibited cells, cellular LIP concentration decreased compared with the control cells (Fig. 3d). Consistent with the decline in LIP, the ferritin H chain expression, which represents the $Fe^{3+}$ content in cell, was increased in these G9a-silenced cells (Supplementary Fig. 5e). Moreover, overexpression of G9a produced the opposite results (Fig. 3e). Meanwhile, overexpression of HEPH in breast cancer cells led to a decrease in cellular iron content, which confirmed the role of HEPH in exporting iron out of cancer cells (Fig. 3f). Encouragingly, a marked decrease in non-heme iron content was also found in G9a-depleted S1 tumor tissue (Supplementary Fig. 2b), which indicates the significance of iron homeostasis in tumor growth. Taken together, these data indicate that G9a loss decreases cellular iron content by increasing the expression and activity of HEPH.

**G9a loss results in reduced iron-dependent cell growth.** We also determined whether the modulation of cellular iron metabolism by HEPH is important for the suppressed growth rate of G9a-silenced cells. Breast cancer cells were grown in media containing the iron chelator desferrioxamine (DFO), which decreased intracellular iron content, or ferric ammonium citrate (FAC), which increased intracellular iron content. Iron overload caused by FAC was associated with an apparent increase in cellular proliferation compared with normal media (Fig. 4a and Supplementary Fig. 3a). In contrast, DFO significantly suppressed cell growth in vitro. Moreover, we found that FAC reversed the diminished proliferation of G9a-silenced cells. Furthermore, DFO increased cell cycle arrest, apoptosis and cell death caused by G9a inhibitor UNC0638 in breast cancer cells, whereas FAC reversed these effects (Figs. 4a–c). As iron depletion in cells increases the level of reactive oxygen species (ROS) leading to DNA damage[22], we also determined whether supplemental iron reduced DNA damage caused by G9a inhibitors using a neutral comet assay. As shown in Supplementary Fig. 3b, the degree of cellular DNA double-strand breaks (DSBs) increased after G9a inhibition in MCF-7 and MDA-MB-231 cells, as evidenced by the frequent appearance and expanding volume of comet tails, as well as the shrinkage of comet heads. DFO exacerbated the damage and increased the expression of γH2AX, which is a marker of DNA DSBs, whereas FAC reduced the effect (Supplementary Fig. 3b, c).

All our observations indicate that the regulation of iron homeostasis is important for G9a-mediated cell survival and proliferation.

**HEPH is a functional target in G9a-promoted proliferation.** We next determined whether HEPH reverses G9a-mediated phenotypes. HEPH has not previously been implicated in cancer-related processes; however, analysis of breast cancer-paired samples in the Ma Breast Statistics from ONCOMINE database showed a significant downregulation of the HEPH transcript in ductal breast carcinoma versus correspondent normal tissues in multiple independent studies (Supplementary Fig. 4b). If the repressive effect of G9a on HEPH expression is important for the growth-promoting functions of G9a, we would expect loss of HEPH to facilitate breast cancer cell survival. Indeed, infection with two HEPH siRNAs significantly reduced the levels of HEPH in MDA-MB-231, MCF-7 and ZR-75-30 cells, meanwhile accelerating cell growth and clonogenic activity in these cell lines (Figs. 4d, e and Supplementary Fig. 4a, 7a), with a concomitant increase of cellular labile iron content (Fig. 4f and Supplementary Fig. 4a). These demonstrated that the decreased HEPH expression is required for proliferation of breast cancer cells. To further confirm the importance of HEPH regulation by G9a in tumorigenesis, we suppressed HEPH expression in G9a-silenced breast cancer cells. As expected, knockdown of HEPH using siRNAs partially restored the intracellular iron concentration and cell growth of G9a-silenced cells (Figs. 4g, h and Supplementary Fig. 7b). Together, these data support the idea that increased HEPH expression induced by G9a loss contributes to decreased proliferation of G9a inhibition.

**HEPH is regulated by G9a in a SET-dependent manner.** We had previously investigated the upregulation of G9a enzymatic-specific inhibitors BIX-01294 and UNC0638 on HEPH expression. To confirm the importance of G9a HMTase activity in repressing HEPH, we transfected G9a knockdown MDA-MB-231 cells with G9a wild-type (G9a WT) or SET domain-deleted (G9a-ΔSET) expression plasmids; HEPH mRNA and protein levels were then evaluated. We found that G9a-ΔSET did not reduce HEPH expression in G9a knockdown cells, as it did in G9a WT cells (Figs. 5a, b), which indicates that G9a-mediated down-regulation of HEPH expression is dependent on its HMTase activity.

Next, we carried out chromatin immunoprecipitation (ChIP) analysis to investigate G9a-mediated transcriptional regulation of HEPH. A series of primers coordinated to the regions in the HEPH promoter were designed for ChIP assays to determine the H3K9 dimethylation and G9a-binding regions of the HEPH promoter in MCF-7 and MDA-MB-231 cells. Four representative regions spanning ~ 2500 bp upstream of the transcription initiation site of the HEPH gene were investigated (Fig. 5c). Pro1 was located far upstream of the HEPH promoter (0.2210 bp) as a negative control, whereas Pro2 and Pro3 were located downstream of the HEPH promoter (0.1250 and 0.450 bp), representing the important regulatory regions of the HEPH gene. We observed decreased G9a recruitment as well as decreased

**Fig. 1** G9a inhibition represses breast cancer cell growth and proliferation in vitro and in vivo. Proliferation assay. **a** Silencing G9a repressed breast cancer cell colony formation ability (*up* panel) and cell growth (*down* panel). Western blotting analysis of G9a depletion in breast cancer cells. **b** Overexpressed G9a in MCF-7 and MDA-MB-231 cells promoted colony formation (*up* panel) and cell growth (*down* panel) in vitro. $n = 3$. **c** G9a loss arrested breast cancer cells in the G1 phase. The cell cycle of G9a knockdown cells or breast cancer cells treated with 5 μM G9a inhibitor UNC0638 for 24 h was investigated by flow cytometry. **d** G9a inhibitors arrested breast cancer cells in G1 phase. Protein levels of c-Myc, Cyclin D1, E2F1 and p21 in G9a inhibitor-treated MCF-7 and MDA-MB-231 cells were tested. **e** Breast cancer cells treated with G9a inhibitors UNC0638 and BIX-01294, and the inhibitory effects on cell proliferation were measured using the Sulforhodamine B (SRB) assay (mean $IC_{50}$ values were calculated from at least three independent experiments). **f** The inhibition of breast tumor growth in vivo of S1 G9a knockdown cells was assessed. $n = 6$. Each bar represents the mean ± SD. Results were representative of three independent experiments, *$P < 0.05$, **$P < 0.01$ and ***$P < 0.001$

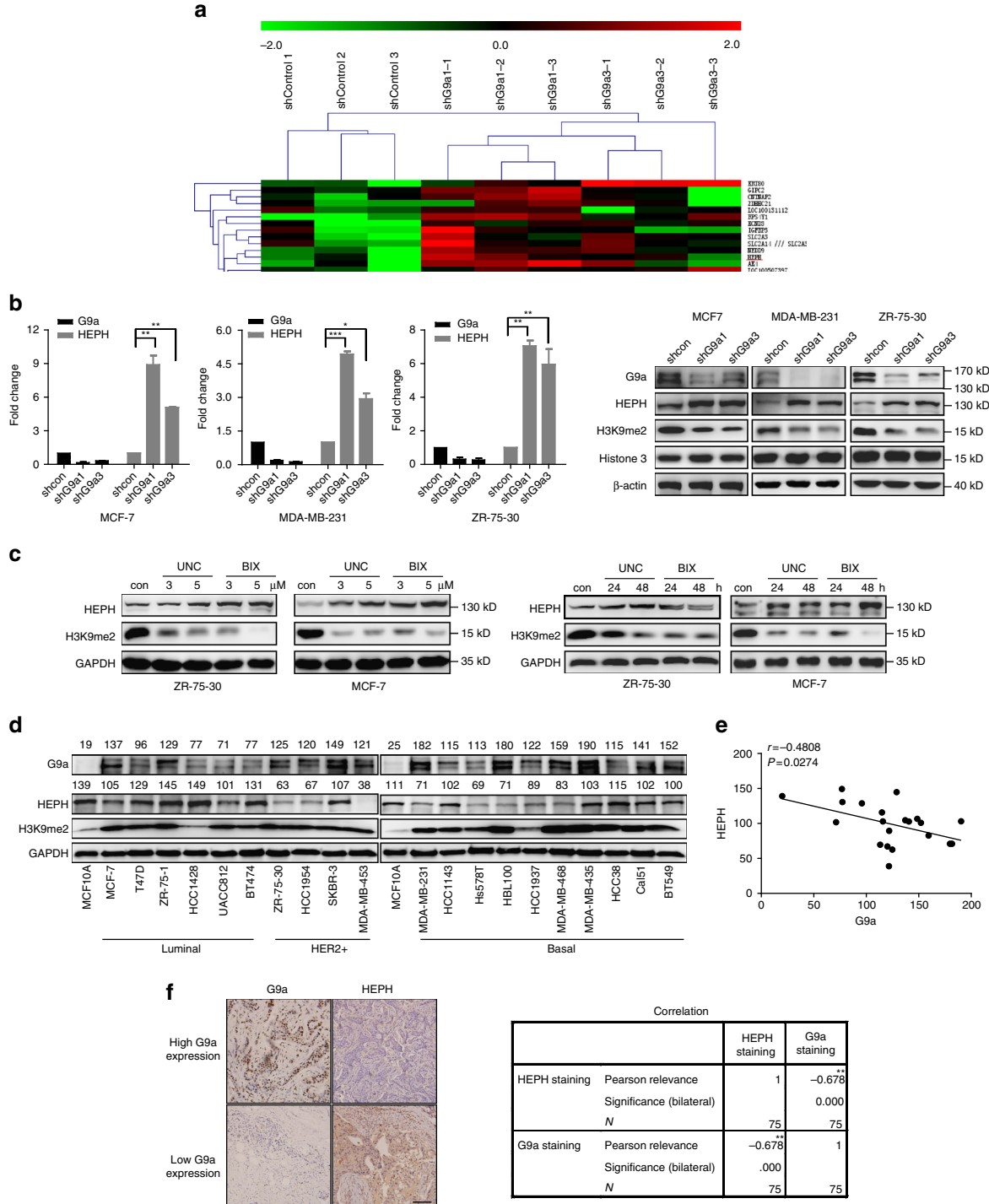

**Fig. 2** G9a negatively regulates HEPH expression. **a** Microarray profiling of gene expression in MDA-MB-231 G9a knockdown cell lines. Heat map values represent the log2 fold change of read counts relative to the counts in the shcontrol cells (*lane 1–3*). Validation of the microarray analysis results by reverse-transcription–PCR and western blotting. Relative HEPH mRNA and protein levels in G9a knockdown **b** and G9a inhibitor-treated **c** cell lines were evaluated. The levels of HEPH and G9a expression in human normal mammary epithelial cell MCF10A and twenty breast cancer cell lines were quantified by Image J **d** and the correlation between them was analyzed using GraphPad; GAPDH was used as a loading control **e**. Data represent the mean of three independent experiments. **f** Immunohistochemical staining analysis of G9a and HEPH proteins in serial sections from breast cancer patients. Pearson's relevance was used to compare G9a and HEPH immunostaining. Note inverse correlation of G9a and HEPH protein expression in tumor cells. Shown are representative sections. *Scale bar*, 50 μm. Results are representative of at least three independent experiments compared with the value for the untreated control and are shown as means with SD. Two-tailed unpaired Student's *T*-test was performed. *$P < 0.05$, **$P < 0.01$ and ***$P < 0.001$

levels of H3K9-me2 only in the *HEPH* promoter 2 region (0.1250–0.870 bp) when G9a was depleted, whereas the over-expressed G9a cell lines gave the opposite result, confirming the possibility of G9a recruitment to the *HEPH* promoter (Figs. 5c, d). The decreased G9a recruitment and reduced levels

of H3K9-me2 on the *HEPH* promoter in G9a knockdown cells could be rescued when G9a WT expression vector was introduced, whereas G9a-ΔSET expression vector could not reverse the reduction (Figs. 5c, d), which indicates that HEPH is regulated by G9a in a SET-dependent manner.

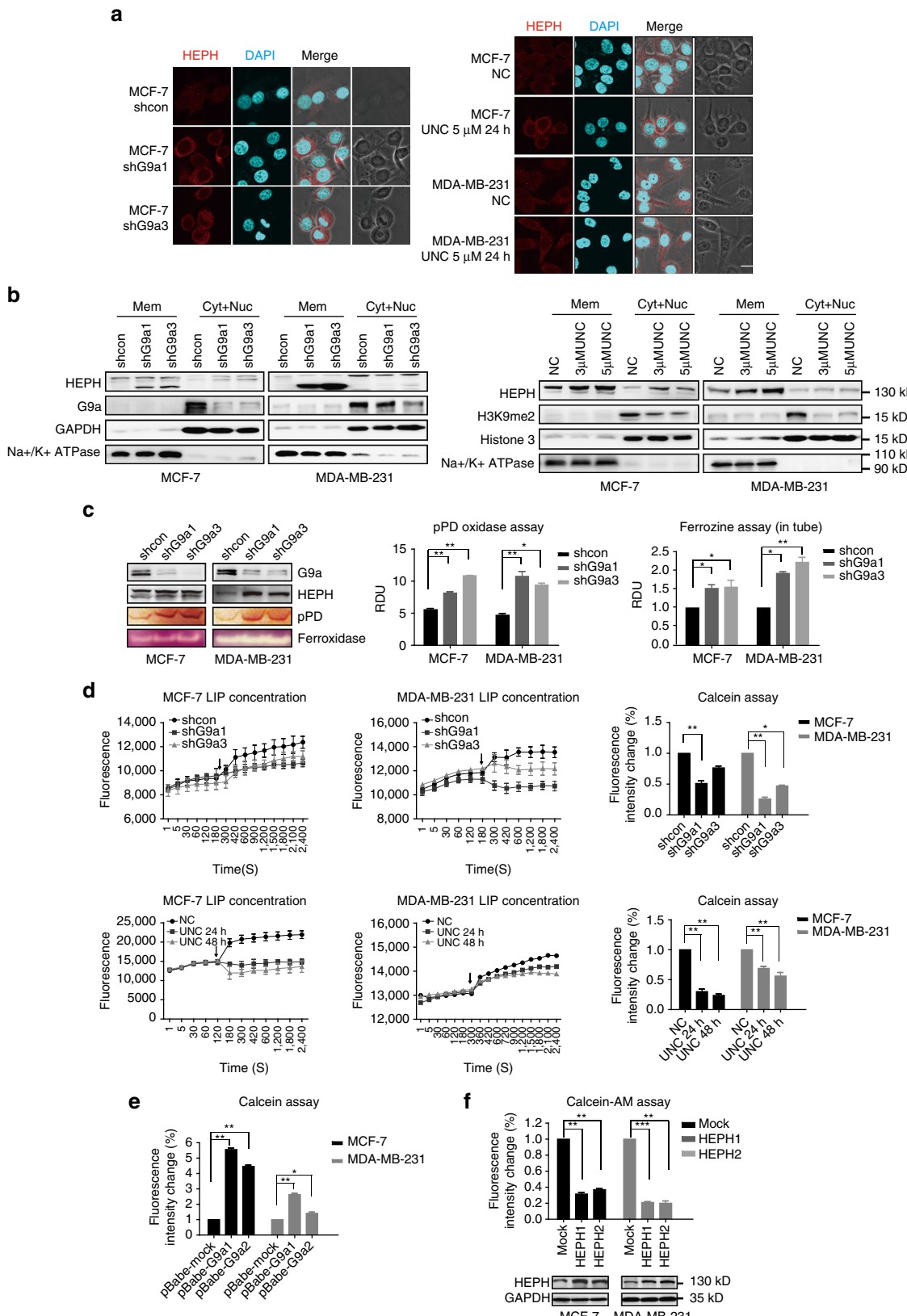

To further understand the mechanisms underlying HEPH transcriptional regulation via G9a, we conducted a promoter reporter assay using a pGL3-*HEPH*-luc reporter system. According to previous reports[23], we cloned three fragments with different *HEPH* promoter lengths and engineered these *HEPH* promoter fragments into pGL3 basic luciferase reporter vectors (Fig. 5e). Consistent with the real-time PCR and western blotting results, HEPH transcription was repressed by G9a overexpression and stimulated by depletion of G9a or UNC0638 in MDA-MB-231 cells in the pGL3-*HEPH*-P1-luc system (Fig. 5e), which suggests that the 0.918–0.366 bp region might be the G9a transcription target of the *HEPH* promoter (referring to the results of the ChIP analysis). We also verified that transcriptional repression of HEPH by G9a is dependent on its HMTase activity using the SET domain-deleted G9a mutant (Fig. 5f). To further confirm the role of G9a for negative regulation of HEPH transcription, we examined the effect of UNC0638 in G9a-overexpressed MDA-MB-231 cells. As expected, G9a-mediated HEPH transcriptional repression was restored by increasing the concentration of UNC0638 (Fig. 5g). Thus, the data described above clearly indicate that G9a negatively regulates HEPH transcription by directly acting on the *HEPH* promoter (0.918–0.870 bp region), which is dependent on HMTase activity.

**G9a interacts with HDAC1 and YY1 to silence HEPH**. We further analyzed the *HEPH* promoter sequence in the 0.918–0.870 bp region, to determine which transcription factors co-regulate HEPH expression with G9a. Among them, we found two YY1 binding sites (0.880 and 0.920 bp sites in the *HEPH* promoter), indicating that YY1 may be involved in HEPH transcription. As a ubiquitous and multi-functional polycomb-group protein family transcription factor, YY1 is able to activate or repress gene expression in different cellular contexts and interacts with a wide variety of regulatory proteins[24–26]. We ascertained whether YY1 has a synergistic effect with G9a on the negative regulation of HEPH expression using real-time PCR and western blotting. Two independent YY1 siRNAs greatly reduced YY1 endogenous protein levels and enhanced HEPH expression alone or in combination with G9a silencing (Fig. 6a and Supplementary Figs 8a, 9). Moreover, co-transfection of G9a and YY1 further repressed HEPH expression, whereas YY1 siRNAs restored the HEPH transcriptional repression induced by G9a overexpression (Fig. 6b). The luciferase reporter assay showed the same result that G9a and YY1 alone each repressed HEPH transcription, whereas co-overexpression greatly inhibited HEPH transcription (Fig. 6c). These data strongly imply that negative regulation of HEPH transcription by G9a is dependent on the presence of YY1. The ChIP assay also indicated that YY1 knockdown by siRNA reduces G9a recruitment and H3K9-me2 abundance in the *HEPH* Pro2 promoter along with YY1 (Fig. 6f).

HDACs also serve with G9a as epigenetic co-repressors to exert repressive gene regulation[27,28]. Therefore, we determined whether HDACs function in G9a- and YY1-induced HEPH inhibition. We studied the effect of overexpression of HDAC subtype members on *HEPH* promoter activity. MDA-MB-231 cells were transfected with *HEPH* promoter reporter, together with pEGFP-hG9a and five human HDACs (HDAC1, 2 and 3 for class I, and HDAC4 and 6 for class II) individually. The results demonstrated that the five tested HDACs exerted distinct effects on the *HEPH* promoter activity, among which only HDAC1 had a much more prominently synergetic effect on G9a-mediated HEPH repression (Fig. 6d). Simultaneously, siRNA-mediated silencing of endogenous HDAC1, but not of HDAC2, restored the decreased protein level of HEPH mediated by overexpressed G9a. Moreover, co-transfection of exogenous G9a and HDAC1 but not HDAC2 further reduced HEPH expression, indicating that HDAC1 really served as another co-repressor with G9a (Fig. 6e and Supplementary Figs 8b, 9). The HDAC1-specific inhibitor MS275 also synergistically increased the protein level of HEPH mediated by G9a inhibition (Fig. 6e). Further ChIP assays were performed again to reconfirm the co-repressors of G9a on the HEPH promotor. The results indicated that G9a, YY1, and HDAC1, but not HDAC2, strongly bond to the HEPH-Pro2 region (Fig. 6g and Supplementary Fig. 5a) and siRNA-mediated depletion of HDAC1 also diminished the abundance of G9a and YY1 on the HEPH promoter region (Supplementary Fig. 5b). The co-immunoprecipitation (Co-IP) results clearly indicated that the three proteins formed a multi-molecular complex with each other (Supplementary Fig. 5c). These data strongly suggest that YY1 and HDAC1 are involved in G9a-mediated HEPH transcriptional repression.

**G9a^high^/HEPH^low^ correlate with poor survival in breast cancer**. These observations prompted us to investigate the relevance of the G9a-HEPH pathway to human disease. As shown previously, elevated expression of G9a in human breast cancer defines a subset of patients with a worse prognosis. It is striking that reduced HEPH levels are also significantly correlated with poor prognosis in the Kaplan–Meier Plotter database (Fig. 7b; the desired Affymetrix ID is valid: 203902_at HEPH; survival curves are plotted for all patients ($n = 4142$)). Moreover, we confirmed the database results in malignant breast tissues from clinical patients by performing tissue microarray analysis on 75 pathologist-verified and clinically annotated breast tumor samples. Patients who with a G9a^high^ tumor had an even worse prognosis in this retrospective analysis (Fig. 7a I), with a median overall survival of 56.9 months compared with 103.4 months in the G9a^low^ tumors. A low level of HEPH was also correlated with overall lower disease-free survival in the same 75 patient samples (Fig. 7a II). Finally, expression of a high level of G9a and a low level of HEPH correlated both with each other and a worse prognosis in these samples (Fig. 7a III). Taken together, our findings were consistent with the model whereby elevated G9a in breast cancer allows aberrant hypermethylation of the HEPH promoter, suppressing HEPH transcription, which then increases intracellular LIP and drives breast tumor progression.

**Discussion**

Interrogation of the published literature reveals that G9a is overexpressed in various tumors, suggesting its oncogenic effects. However, the link between G9a and carcinogenesis remains

---

**Fig. 3** Upregulated HEPH with high ferroxidase activity accumulates on the cellular membrane and leads to decreased LIP. **a** Immunofluorescence staining analysis of HEPH proteins in breast cancer cells transfected with G9a shRNA or treated with 5 µM UNC0638 for 24 h. Shown are representative sections. *Scale bars*, 10 µm. **b** Western blotting tested HEPH level in separated cell components of membrane and cytoplasm. **c** HEPH activity in G9a knockdown cells was measured by the oxidation of pPD and ferrozine assays. The homologue ceruloplasmin served as a positive control. **d** The cellular labile iron pool in G9a knockdown or inhibited cells was measured using the calcein-AM assay. The *arrows* indicate when the iron chelator was added. **e** The cellular labile iron pool in G9a-overexpressed cells was measured. **f** Western blotting tested HEPH overexpression in MCF-7 and MDA-MB-231 cells and the cellular labile iron pool in these cells were measured. All the results are presented as means ± SD from three independent experiments. Two-tailed unpaired Student's *T*-test was performed. *$P < 0.05$, **$P < 0.01$ and ***$P < 0.001$

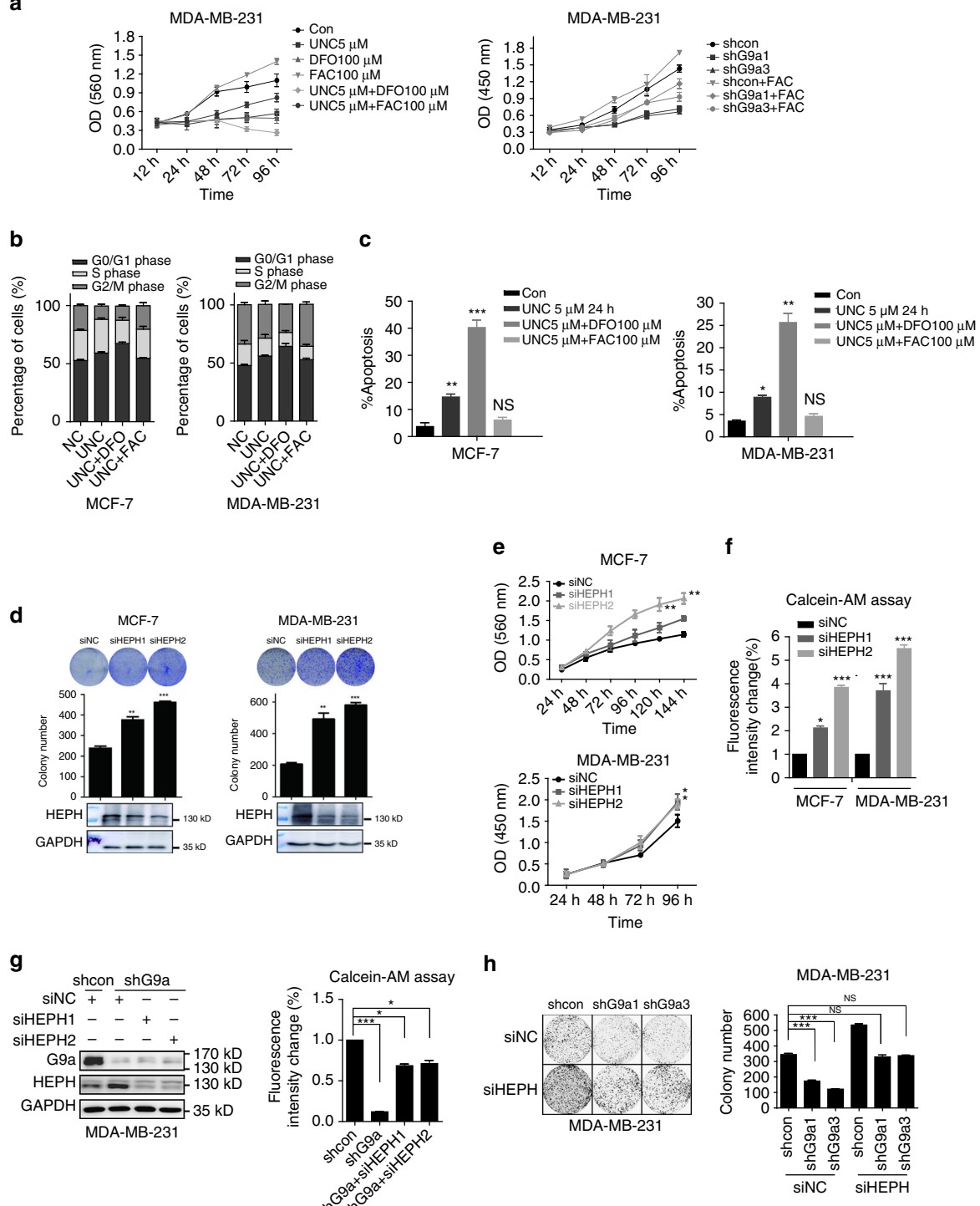

**Fig. 4** G9a regulates breast cancer growth through cellular iron content. **a** Growth curves of MDA-MB-231 cells with or without G9a inhibitor treatment and G9a knockdown cell lines cultured in standard media or media supplemented with 100 μM DFO or 100 μM ferric ammonium citrate (FAC). Cell cycle arrest **b** and apoptosis **c** caused by G9a inhibitor could be enhanced or reduced under iron depletion or overload. Data represents the mean of three independent experiments. Silencing HEPH promoted MCF-7 and MDA-MB-231 colony formation **d** and cell growth **e**. **f** The cellular labile iron pool (LIP) in HEPH knockdown cells were tested. **g** Western blotting analysis of G9a and HEPH expression levels (*left*) and cellular LIP content of MDA-MB-231 cells after G9a and/or HEPH knockdown (*right*). **h** Colony formation ability was restored when HEPH was depleted by siRNA in G9a knockdown cells. The results are presented as means ± SD from three independent experiments. Two-tailed unpaired Student's *T*-test was performed. *$P < 0.05$, **$P < 0.01$ and ***$P < 0.001$, *NS* not significant

poorly understood. In this study, we demonstrated that G9a exerts its oncogenic function by destroying cellular iron homeostasis. This histone methyltransferase regulates cellular iron metabolism through HEPH, with important implications for breast tumor cell growth. We found that enhanced iron content

and decreased HEPH expression are required for the increased proliferation of G9a-overexpressed breast cancer cells in vitro and in vivo. This idea was further validated by the finding that depletion of G9a stimulates HEPH expression and activity, and leads to decreased iron content, which suppresses the

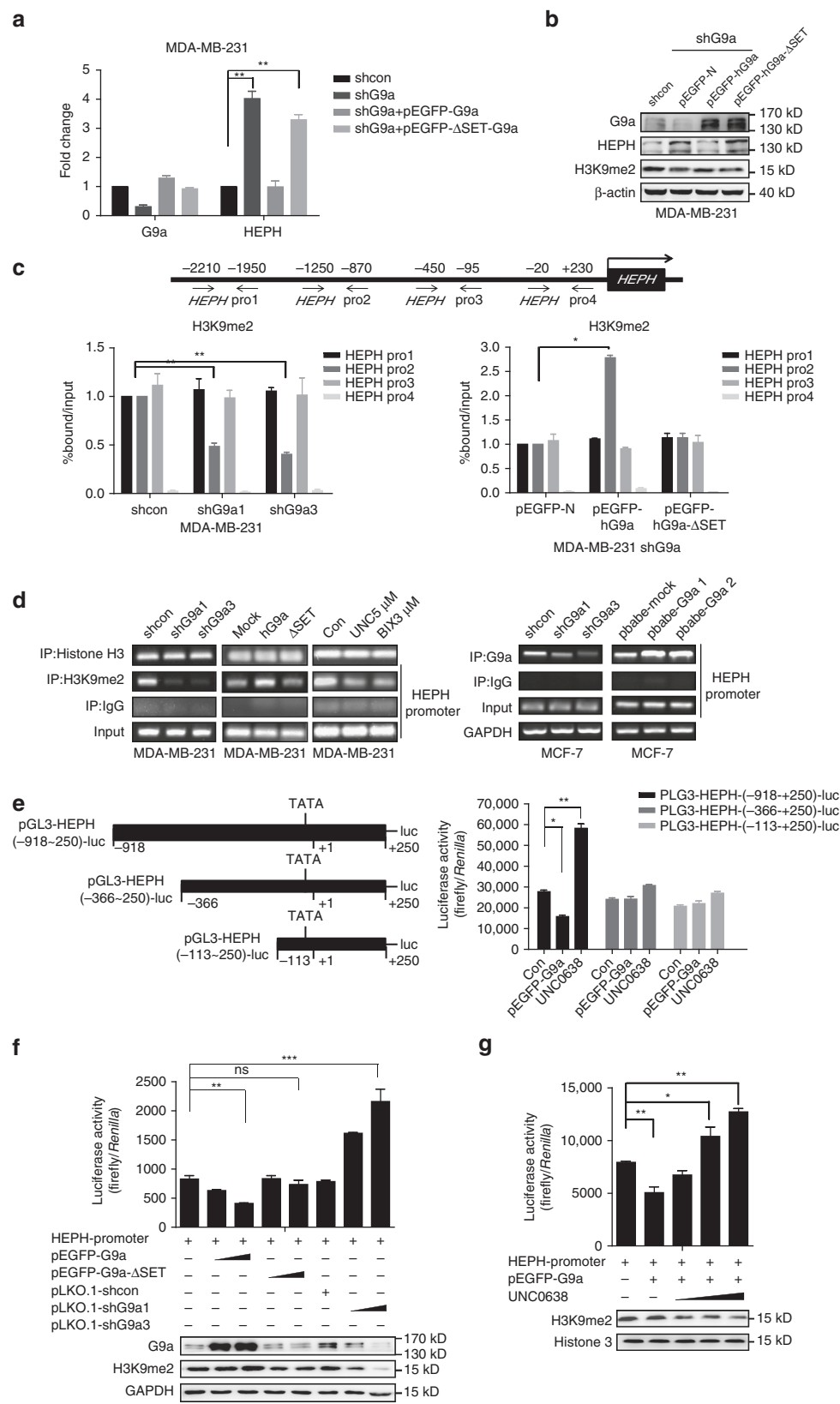

proliferation of breast cancer cells. Further, our data demonstrate a traditional role of G9a as a transcriptional repressor that co-exists with YY1 and HDAC1, and contributes to the reduction of HEPH expression. In addition, another important finding is that high G9a and low HEPH are associated with poor prognosis in breast cancer patients. Thus, our observations raise the exciting possibility that G9a and HEPH are potential prognostic markers of breast cancer progression and targets for therapeutic intervention.

As a fundamental trace element involved in cell metabolism, division and proliferation, iron has also been implicated as an important factor in cancer development[29–31]. Many cancers exhibit an increased requirement for iron, presumably because of the need for iron as a cofactor in proteins that are essential to sustain growth and proliferation[32,33]. Population-based studies have taken a general approach to examine the relationship between iron and cancer risk. Although the results are not always consistent, the studies collectively support a model in which increased levels of iron in the body are associated with increased cancer risk[9,34–36]. Cancer cells always differ from their non-malignant counterparts in the levels or activity of many of the proteins that are involved in iron metabolism. From the cell biology perspective, it is now well accepted that the malignant state in breast epithelial cells is characterized by a deregulation in cellular iron homeostasis, as revealed by differences in the expression of several iron-associated proteins related to markers of poor outcome[37–40]. Lamy et al.[37] demonstrated that in breast cancer cells, the expression/activity of several iron-related proteins, such as ferritin, hepcidin, and FPN (also known as Ireg1), are deregulated and that these alterations may have a prognostic impact on patients with breast cancer. Decreased levels of FPN, which is the only iron efflux pump in vertebrates, are associated with ascending levels of the LIP in cultured breast cancer cells, and increased growth of breast tumor xenografts. Moreover, low FPN expression was significantly associated with a poor prognosis in four separate cohorts comprising approximately 800 patients with breast cancer[12]. Transferrin receptor 1 (a), a cell surface receptor that is responsible for transferrin-mediated iron uptake, is highly expressed in many cancers, including breast cancer[38,41]. Consequently, TfR1 antibodies have been used to inhibit tumor growth[42]. In this study, we indicated that HEPH, another important iron-associated protein that makes a substantial contribution to the regulation of cellular iron levels, has a key role in the clinical behavior of breast cancer. Currently, HEPH is only known to play an important role in the intestine, eye and brain, with cells in these tissues accumulating iron when HEPH expression is perturbed[23]. We proved that HEPH is downregulated by G9a in breast cancer cell lines and in human breast cancer samples. We also observed for the first time that decreased HEPH is associated with increased levels of the LIP in cultured breast cancer cells and with the stimulated growth of breast cancer in vitro and in vivo. The most important observation is that decreased HEPH expression is significantly associated with a poor prognosis in breast cancer, and the combination of high G9a and low HEPH is associated with shorter survival times. All these factors indicate that the measurement of G9a and HEPH levels in breast tumors could be useful in breast cancer prognosis. We also investigated whether any of the other iron metabolism-related proteins mentioned above had any effect on G9a function. We found hardly any significant regulatory relationship between G9a and these proteins, except for HEPH. The mRNA and protein levels of DNMT1, FPN and TfR1 had no significant effect according to our study (Supplementary Fig. 5d, e). These data suggest that altered HEPH or iron levels may play a previously unappreciated role in breast cancer behavior, although additional investigation is required to confirm this. Here we also showed a novel connection between a histone methyltransferase and cellular iron metabolism. We observed that over-expressed G9a results in iron accumulation in breast cancer cells and stimulates cell growth in vitro and in vivo. These data reveal a mechanism by which G9a regulates tumor growth by manipulating cellular iron homeostasis in breast cancer development.

Our current work also highlights the detailed mechanism of HEPH expression regulated by G9a. Published reports suggested a potential role for G9a in human cancers via negative regulation of UHRF1 and JAK2 transcription in leukemia[28], or via methylation of the non-histone protein p53[43]. In the present study, we investigated the novel G9a target gene *HEPH* in breast cancer. We proved that G9a operates as a negative regulator of *HEPH* expression via YY1 and HDAC1 interaction, and is recruited to the *HEPH* promoter during breast cancer cell growth. Identifying these key repressive molecules that are responsible for G9a-mediated transcriptional repression of HEPH is important for a better understanding of complicated epigenetic regulation during breast cancer progression.

In summary, we demonstrated that G9a is involved in iron metabolism by modulating HEPH expression. We propose that G9a has an upstream regulatory role in HEPH-mediated cellular iron homeostasis leading to iron accumulation and stimulates breast cancer progression through its epigenetic silencing machinery. Thus, it will be interesting to examine the role of G9a in systemic iron homeostasis and iron-related human diseases. Our molecular model revealed a new insight of epigenetics that regulates tumor growth by manipulating cellular iron homeostasis. Whether other epigenetic players exist that participate in the process and have similar underlying patterns requires more investigation.

**Fig. 5** G9a-mediated transcriptional repression of HEPH is HMTase-dependent. Relative HEPH mRNA **a** and protein levels **b** of HEPH in G9a knockdown, G9a WT, and G9a SET domain deleted rescued MDA-MB-231 cells. **c** Schematic diagram of primer pairs of the human *HEPH* promoter region (GeneBank accession number: NC_000023.11) (*upper* panel) in the ChIP assay and real-time PCR analysis (*lower* panel). MDA-MB-231 G9a knockdown cells or cells transfected with pEGFP-G9a or pEGFP-G9a-ΔSET were harvested and analyzed (mock as control). Cross-linked samples were immunoprecipitated with anti-H3K9-me2 and anti-G9a antibody, and the precipitated DNA fragments were subjected to real-time PCR in the *HEPH* promoter regions. **d** ChIP immunoprecipitation of recruitment of G9a to the *HEPH* promoter region was normalized by input. **e** Schematic representation of the different lengths of the *HEPH* promoters, which contain various putative G9a-binding sequences, constructed to form the pGL3 luciferase vector. The "+1" represents the transcription start site. *HEPH* promoter activities in MDA-MB-231 cells were weakened by pEGFP-hG9a or strengthened by treatment with 5 μM UNC0638. **f** MDA-MB-231 cells were co-transfected with the pGL3-*HEPH* promoter (0.5 μg) and pEGFP-G9a (0.5 and 1 μg), pEGFP-G9a-ΔSET (0.5 and 1 μg), pLKO.1 (1 μg) as shcon and G9a shRNAs (0.5 and 1 μg), along with the TK-*Renilla* luciferase expression plasmid (pRL-SV-luciferase vector). Cell extracts were assayed for luciferase activity. G9a overexpression or knockdown was confirmed by western blotting analysis. **g** Restoration of G9a-mediated *HEPH* transcriptional repression by UNC0638. The pGL3-*HEPH* promoter (0.5 μg) and pEGFP-G9a (1 μg) were co-transfected into MDA-MB-231 cells. Twenty-four hours after transfection, UNC0638 (1, 3, and 5 μM) was supplied for 24 h and the luciferase activity was measured. Firefly luciferase activity levels were normalized to those of the *Renilla* luciferases. The pEGFP empty vector was used as a negative control and was added to maintain equal amounts of total transfected DNA. All data are representative of at least three independent experiments and are presented as means ± SD. Two-tailed unpaired Student's *T*-test was performed. *$P < 0.05$, **$P < 0.01$ and ***$P < 0.001$, NS not significant

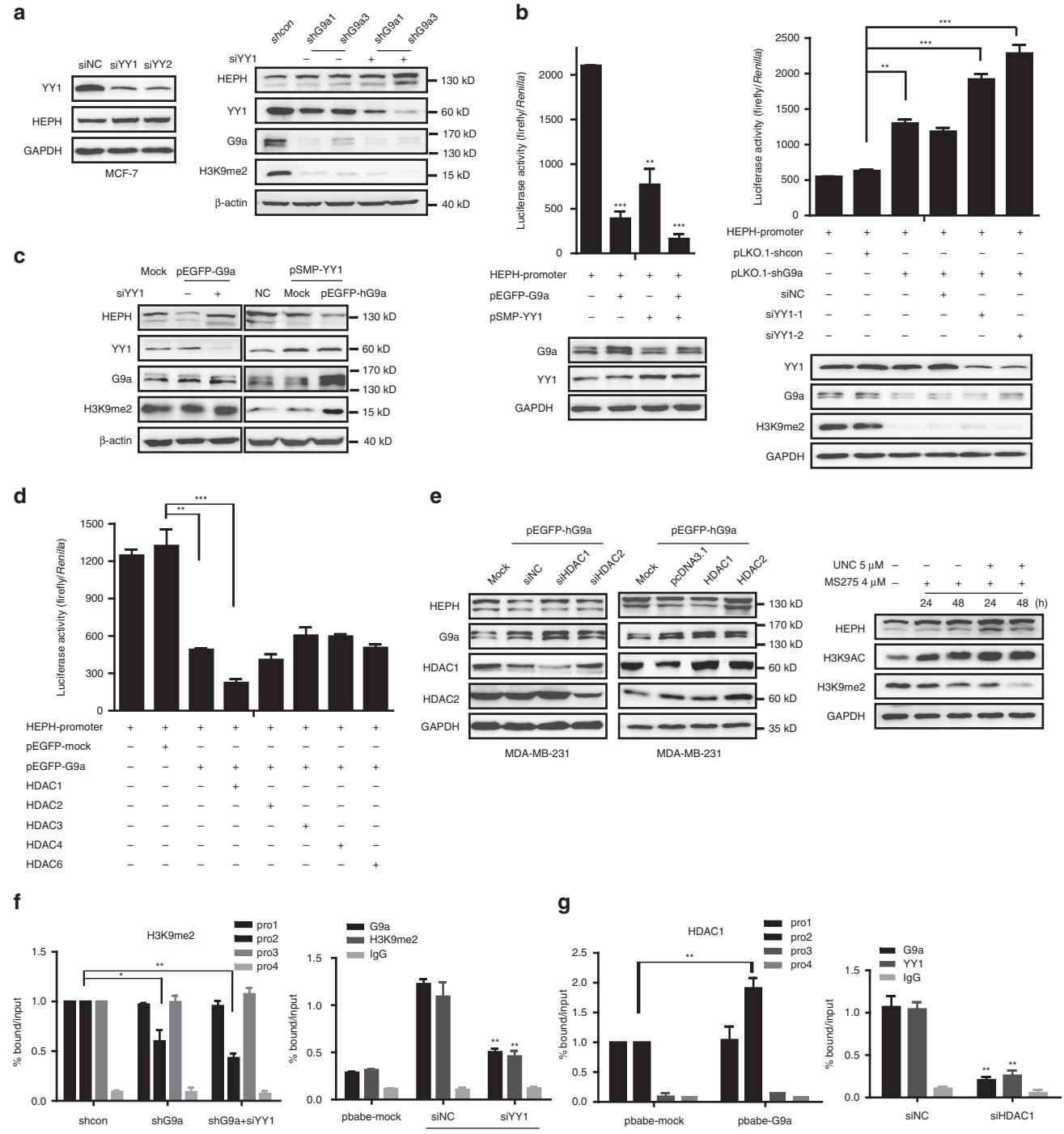

**Fig. 6** G9a silences the expression of HEPH via assembling a co-repressor complex with YY1 and HDAC1. **a–c** MCF-7 cells were transfected with two independent YY1 siRNAs. After 48 h, HEPH protein and mRNA levels and *HEPH* promoter luciferase activity were examined. Expression of the transfected constructs is shown in the western blotting analysis. **d** pGL3-*HEPH* promoter and the indicated constructs were co-transfected into MDA-MB-231 cells. Twenty-four hours after transfection, cell extracts were assayed for luciferase activity. **e** Silencing and overexpression of HDAC1, but not HDAC2, contributed to the upregulation or downregulation of HEPH mRNA and protein level, respectively. The HDAC1-specific inhibitor MS275 was synergetic with UNC0638 in increasing HEPH expression in a time-dependent manner. **f**, **g** The abundance of H3K9-me2 and the binding levels of G9a and HDAC1 in the Pro2 region of the *HEPH* promoter were determined by ChIP in G9a knockdown or overexpressed cells treated with siYY1. The results are presented as means ± SD from three independent experiments. Two-tailed unpaired Student's *T*-test was performed. *$P < 0.05$, **$P < 0.01$ and ***$P < 0.001$

## Methods

**Chemicals and antibodies**. UNC0638 (#U4885) and BIX-01294 (#B9311) were both purchased from Sigma (St. Louis, MO, USA). The iron chelator DFO was from Novaritis (Switzerland). FAC was from JK Chemical (#F5879, Shanghai, China). Kits of the membrane and cytosol protein extraction kit (#P0033), the ROS assay (#S0033) and the cell lysis buffer radio-immunoprecipitation assay (#P0013B)

were purchased from Beyotime (Nantong, China). Monoclonal antibodies specific for G9a (1:1,000; #3306), YY1 (1:1,000; #2185), HDAC1 (1:1,000; #34589), HDAC2 (1:1,000; #57156), Histone H3 (1:2,000; #4499), K9 dimethylated histone H3 (1:2,000; #4658), β-actin (1:5,000; #8457) and glyceraldehyde 3-phosphate dehydrogenase (GAPDH) (1:5,000; #5174) were purchased from Cell Signaling Inc. (Danvers, MA, USA). The other antibodies were as follows: HEPH antibody (1:500;

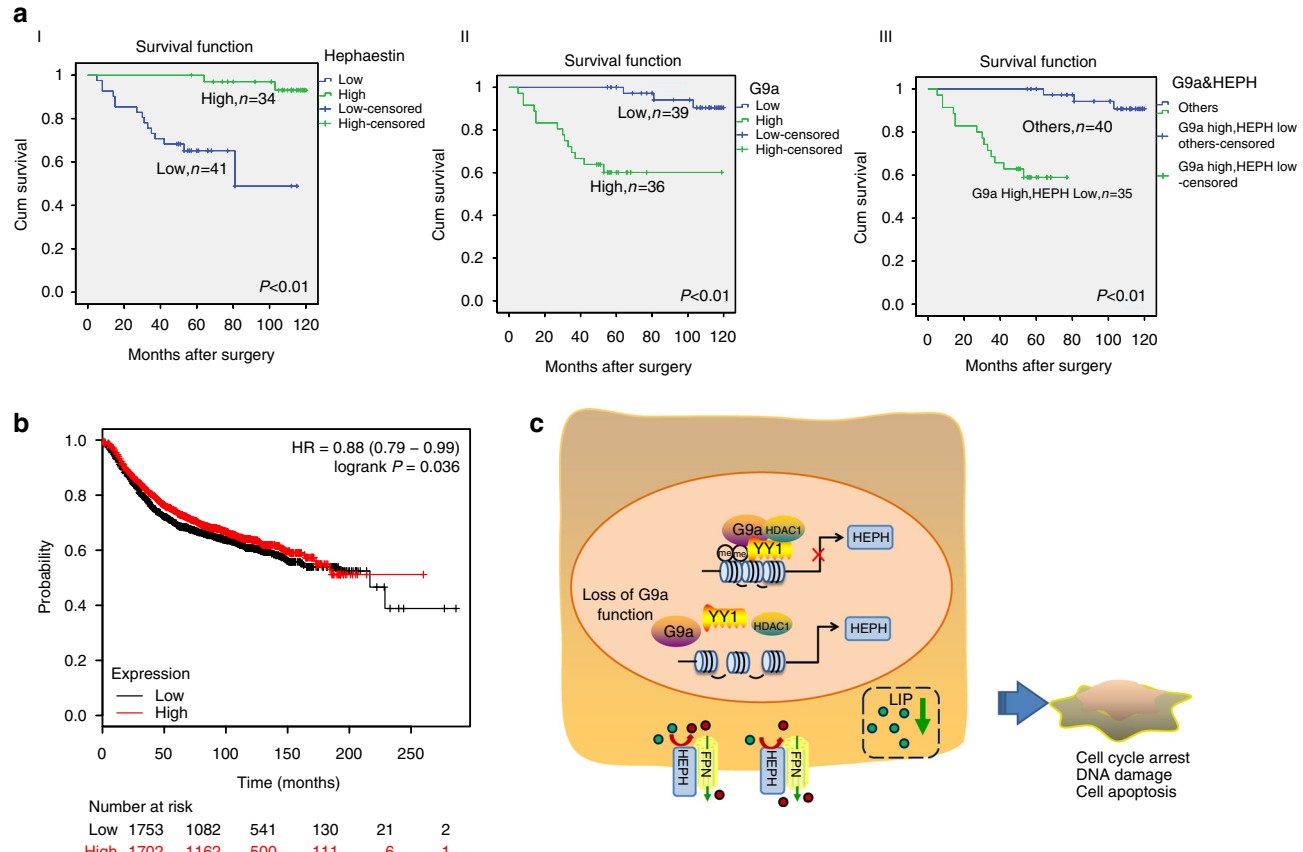

**Fig. 7** High levels of G9a and low levels of HEPH correlate with poor survival in breast cancer. **a** G9a and HEPH prognostic interactions. Associations between OS and high or low G9a and HEPH expression levels (based on mean partitioning) in a combined multi-institutional population-based cohort consisting of 75 breast cancer cases. Kaplan–Meier plots and log-rank *P*-values are shown for (I) HEPH expression, (II) G9a expression, (III) high G9a dichotomized by low HEPH. **b** Public breast cancer database (KM-Plotter) was queried to examine the association between patients with breast cancer RFS and HEPH expression, the log-rank test *P*-value was indicated. **c** Schematic diagram depicting the regulation of HEPH in breast tumor cells. G9a as an HMTase activity-dependent repressor collaborates in the complex with YY1 and HDAC1, and works coordinately to contribute to the reduction of HEPH expression. Silencing of G9a upregulates HEPH, inhibits breast cancer cell proliferation and cell survival via upregulation of HEPH transcription and induces HEPH-mediated iron homeostasis disruption upon greater iron export. *Green* particles represent ferrous iron; *red* particles represent ferric iron

#sc-365365, Sigma), green fluorescent protein (GFP) antibody (1:1,000; #sc-69779, Santa Cruz Biotechnology), Alexa Fluor 488 goat anti-mouse IgG (H + L) antibody (1:200; #A-11001, Life Technologies, OR, USA) and Alexa Fluor 594 goat anti-rabbit IgG (H + L) antibody (1:200; # A-11012, Life Technologies).

**Cell culture**. Commercialized breast cancer cell lines MCF-7, MDA-MB-231, MDA-MB-468, MDA-MB-435, ZR-75-30 and T47D were purchased from the American Type Culture Collection (Manassas, VA, USA). The S1 and HBL100 cell lines were obtained from cell bank of the Chinese Academy of Sciences. The SK-BR-3 cell line was purchased from Henlius Biotech, Inc. (Shanghai, China). Cell lines ZR-75-30 and T47D were cultured in RPMI-1640 medium (Gibco, Grand Island, NY, USA); cell lines MCF-7, S1 and HBL100 were cultured in low-glucose Dulbecco's modified Eagle's medium (Gibco); cell line SK-BR-3 was cultured in McCoy'5A medium (Sigma); and cell lines MDA-MB-231, MDA-MB-468 and MDA-MB-435 were cultured in L-15 medium (Gibco) supplemented with 10–15% heat-inactivated fetal calf serum (Gibco) and 100 U ml⁻¹ penicillin and 100 µg ml⁻¹ streptomycin. All cells were authenticated by the analysis of short tandem repeat profiles and 100% matched the standard cell lines in the ATCC and DSMZ (Deutsche Sammlung von Mikroorganismen und Zellkulturen GmbH) data bank. All cells were tested negative for cross-contamination of other human cells and mycoplasma contamination.

**Plasmids and transfection**. The PLKO.1-shG9a plasmids were generously provided by Dr Jin Jian (University of North Carolina, USA). The pEGFP-hG9a (Addgene ID 330025) and pEGFP-ΔSET-hG9a (Addgene ID 330026) plasmids were obtained from Addgene, and recombined into pLEX to construct pLEX-hG9a and pLEX-ΔSET-hG9a. These plasmids were transfected into 293FT cells with packaging mix pCMV-dR8.2 dvpr and pCMV-VSVG to produce lentiviruses. The stable knockdown and overexpressed G9a cell lines were established as outlined in the Addgene protocols. The HDAC1, 2, 3, 4, 6-Flag plasmids (Addgene ID 13820,

68117, 13819, 13821, 13823) and pSMP-YY1 plasmid (Addgene ID 36357) were also obtained from Addgene. siRNA and plasmid transfections were carried out using Lipofectamine RNAiMax (#13778500) and Lipofectamine 2000 (#11668019, Invitrogen, Carlsbad, CA, USA) respectively, according to the manufacturer's instructions.

**Real-time reverse-transcription PCR**. Total cellular RNA was isolated with TRIzol (#15596018, Invitrogen) and reverse transcribed into complementary DNA using the PrimeScript RT reagent Kit (#RR036A, Takara, Otsu, Shiga, Japan). Real-time reverse-transcription PCR was carried out on an Applied Biosystems 7500 apparatus using SYBR-Green Master mix (#RR820B, Takara) with the following primers: *G9a* 5′-gccaggccgggaggccctggaa-3′ (sense), 5′-ctccagcctgcagcag-cacatg-3′ (antisense); *HEPH* 5′-atgcactgccatgtgactga-3′ (sense), 5′-cttggtgatgacggt-gagg-3′ (antisense); and *GAPDH* 5′-gcaaattccatggcaccgtc-3′ (sense), 5′-tcgccccac ttgattttg-3′ (antisense). The other primer sequences were listed in Supplementary Table 1. The reaction parameters were: 95 °C for 10 min, followed by 42 cycles of 95 °C for 5 s, and 60 °C for 34 s. All samples including the template controls were assayed in triplicate. The relative number of target transcripts was normalized to the number of human *GAPDH* transcripts found in the same sample. The relative quantification of target gene expression was performed with the standard curve or comparative cycle threshold method.

**Colony formation assay and cell proliferation assay**. MCF-7 and MDA-MB-231 stable knockdown cell lines were seeded onto six-well plates at a density of 1,000 cells per well. The cells were cultured for 10–15 days until the colonies became visible. The colonies were fixed in 10% formaldehyde and 10% acetic acid at room temperature for 15 min, and then stained with 1% Crystal Violet (#C6158, Sigma). The cytotoxicity of UNC0638 and BIX-01294 were investigated using a panel of human breast tumor cell lines. Cells plated onto 96-well plates were treated with gradient concentrations of the compounds at 37 °C

for 72 h. A cell proliferation assay was carried out using Sulforhodamine B (SRB; #230162, Sigma).

**Flow cytometry.** Cells were stained with Annexin V–fluorescein isothiocyanate (FITC) and propidium iodide (PI), and then evaluated for apoptosis by flow cytometry according to the manufacturer's protocol (#V13242, Invitrogen). Briefly, after treatment the cells were collected by centrifugation. The cell pellets were suspended in 500 µl of binding buffer and incubated with 5 µl of Annexin V-FITC and 5 µl of a PI solution at room temperature for 15 min. Annexin V and PI staining was measured by flow cytometry on a FACSCalibur instrument (BD Biosciences, NJ, USA) followed by data analysis using FlowJo software.

**Immunofluorescence.** The cells were grown on chamber slides, fixed with 4% paraformaldehyde and permeabilized with phosphate-buffered saline (PBS) containing 0.1% Triton X-100. After blocking with 3% bovine serum albumin (BSA) for 1 h, the cells were incubated with primary antibodies overnight. These cells were then washed three times with PBS, and incubated with Alexa Fluor 488 goat anti-mouse IgG (H + L) or Alexa Fluor 594 goat anti-rabbit IgG (H + L) secondary antibodies. Nuclei were visualized using 4′,6-diamidino-2-phenylindole staining. The fluorescence signals were analyzed using an Olympus Fluor view 1000 confocal microscope.

**Co-IP and ChIP assay.** The 293T and MDA-MB-231 cells were co-precipitated, as described previously[5]. Total cell extracts were precleared with 30 µl of protein A-agarose at 4 °C for 1 h. The supernatant was incubated with the anti-G9a or anti-YY1 with gentle shaking overnight at 4 °C, followed by the addition of 40 µl of protein A/G-agarose beads for another 4 h. The beads were wached and resuspended in 30 µl of 2 × loading buffer and boiled for 10 min. The proteins were separated by SDS–PAGE (10% SDS) and transferred to a nitrocellulose membrane for immunoblot detection with anti-YY1 antibody or anti-HDAC1 antibody.

ChIP assays were carried out according to the manufacturer's protocol (#9005s, Cell Signaling). Briefly, the cells were collected and subsequently cross-linked with 1% formaldehyde. After centrifugation, the resulting pellets were sonicated and the chromatin solution was precleared with 50 µl of ChIP-Grade protein G magnetic beads (Cell Signaling). The soluble fraction was collected and the chromatins were incubated with 5 µl of anti-K9 dimethylated histone H3, anti-histone H3, anti-G9a or anti-YY1 (Cell Signaling) at 4 °C overnight. The CHIP-enriched DNA was analyzed by quantitative PCR using the specific primers described in Supplementary Table 1–3. The enrichment of specific genomic regions were assessed relative to the input DNA followed by normalization to the respective control IgG values.

**Subcellular fractionation.** The cytosolic and solubilized particulate membrane fractions were prepared as described in the Beyotime protocol. All steps were performed at 4 °C. Briefly, the cancer cells were homogenized using a tissue grinder in buffer A (0.025 M Tris-HCl, pH 7.4, 0.025 M NaCl, plus protease inhibitor cocktail) and centrifuged at 16,000 g for 15 min. The cytosolic fractions were obtained by re-centrifuging the supernatants at 10,000 g for 1 h. The pellets were resuspended in buffer B (buffer 1 with 0.25% [v/v] Tween-20), sonicated for 3 × 10 s at 25 Watts in ice water slurry with 15 s chilling in between and re-centrifuged at 16,000 g for 30 min. These supernatants were termed the solubilized membrane fraction.

**Measurement of intracellular calcein-chelatable iron.** The amount of calcein-chelatable iron within both the control cells and the cells initially exposed to G9a inhibitors were assayed along the G9a knockdown cell lines described previously[44]. Briefly, the treated cells were incubated with 0.15 µM calcein-AM (#C3099, Invitrogen) for 10 min at 37 °C in PBS containing 1 mg ml$^{-1}$ BSA and 20 mM 4-(2-hydroxyethyl)-1- piperazineethanesulfonic acid (pH 7.3). After calcein loading, the cells were trypsinized, washed, re-suspended in the buffer mentioned above without calcein-AM, and placed in 96-cell plates; the fluorescence was monitored ($\lambda_{ex}$ 488 nm; $\lambda_{em}$ 518 nm). Calcein-loaded cells show a fluorescence component ($\Delta F$) that is quenched by intracellular iron. This iron-induced quenching was minimized by the addition of 100 µM DFO, a lipophilic, highly specific and membrane-permeable iron chelator. Cell viability (assayed as Trypan Blue dye exclusion) was >95% and did not change during the assay.

**pPD oxidase and ferroxidase activity assay.** The oxidase activity of HEPH was determined using lysates of breast cancer cells prepared as described previously[20]. Briefly, the cells were washed and lysed in PBS containing 1.5% Triton X-100. The cell homogenates were centrifuged at 13,000 g for 30 min to remove unlysed cells and nuclei. The clear lysates were applied to a native, non-reducing, non-denaturing 10% Tris-glycine polyacrylamide electrophoresis gel. The gels were then incubated with 0.1% pPD (#78429, Sigma) in 0.1 M acetate buffer, pH 5.45, for 2 h and air-dried in the dark. Purified human CP (#C4519, Vital Products, Sigma) was used as a positive control.

The ferroxidase-specific assay differs from the pPD gel assay only in the final assay step[45]. The gels were placed in a fresh solution of 0.00784% Fe (NH$_4$)$_2$(SO$_4$)$_2$·6H$_2$O in 100 mmol l$^{-1}$ sodium acetate, pH 5.0, for 2 h at 37 °C. The gels were then rehydrated with 15 mmol l$^{-1}$ ferrozine solution in the dark. Color development was monitored continuously and quantified by scanning densitometry. CP activity was detected and served as a positive control. For the in-tube assay, cell extracts were incubated with 0.00784% Fe(NH$_4$)$_2$(SO$_4$)$_2$·6H$_2$O substrate in 100 mmol l$^{-1}$ sodium acetate, pH 5.0, for 2 h at 37 °C in the dark. The assay solutions were then rehydrated with 15 mmol/l ferrozine solution in the dark for 30 min. The absorbance of the assay solution was determined in a spectrophotometer at 562 nm. The results are expressed as means ± SD.

**Luciferase reporter assays.** A human *HEPH* promoter reporter (pGL3-*HEPH*-Luc) was constructed as follows: three different portions of the *HEPH* gene proximal region were amplified from human genomic DNA using the primers described in Supplementary Table 3. The resultant amplicons were digested with *Mlu*I and *Xho*I at the primer-encoded restriction sites, and subsequently subcloned into pGL3-Basic (Promega). For the transcriptional activity assay, MDA-MB-231 or 293T cells were seeded into 12-well plates and transfected with the pGL3-*HEPH* promoter (−918/−366/−113 to +250) reporter plasmids, either in the presence or absence of the indicated expression plasmid mentioned above, using Lipofectamine 2000 (Invitrogen). After 24 h, the cells were collected and subjected to a Dual Luciferase Reporter Assay System (# E1910, Promega). The activity of the co-transfected TK-*Renilla* luciferase plasmid was used as a transfection efficiency indicator to normalize the firefly luciferase. Extracts from at least three independent transfection experiments were assayed in triplicate. The results are shown as means ± SD.

**Patients and tumor tissues.** Seventy-five formalin-fixed, paraffin-embedded primary breast tumor tissue samples obtained from (#BR150S01, #BR150S02) Zuo Cheng Biological Technology LTD, Shanghai, China, and their associated clinicopathological information were collected from patients who received surgical resection between 2005 and 2014. None of the patients had received adjuvant therapies before surgery. The tumor specimens were analyzed for G9a and HEPH protein expression. Semi-quantitative immunohistochemistry detection was used to determine the protein levels. We multiplied the positive percentage score by the staining intensity score using the H-score (histochemical score) method analysis considering the tumour component only. After scoring, the data were analyzed by Pearson's $\chi^2$-test and Kaplan–Meier survival analysis.

**In vivo study.** Female thymic BALB/c nude mice, 4–6 weeks old, were housed and maintained under specific pathogen-free conditions with a 12 h light/dark cycle at 25 ± 1 °C, and received food and water ad libitum. All experiments were carried out according to the institutional ethical guidelines on animal care and were approved by the Institute of Animal Care and Use Committee at the Shanghai Institute of Materia Medica (No. 2016-04-DJ-21). We used random number tables as our randomized method to determine the animals allocated to experimental groups. S1 cells with different levels of G9a (G9a shcon, G9a sh1 and G9a sh3) were subcutaneously injected into the right plate of nude mice at a concentration of $5 \times 10^6$ cells/mouse (six mice per group). Tumor diameters were measured two times per week and tumor volumes (V) were calculated using the formula: V = ½ × length × width$^2$.

**Statistical analysis.** Means, SD and SEM were analyzed using Graphpad. Two-tailed Student's $t$-test, two-way analysis of variance (ANOVA) or one-way ANOVA with Dunnett's multiple comparisons test were used to compare the statistical difference between indicated groups. Statistical significance was accepted for $P$-values of <0.05.

**Data availability.** The data that support the findings of this study are available within the article, its Supplementary Information files and from the corresponding author upon reasonable request.

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

## Acknowledgements

We thank Professor Jian Jin (University of North Carolina at Chapel Hill) for the gifts of the plasmids pLKO.1-shControl and pLKO.1-shG9a. We are grateful to Martin J. Walsh for the plasmids pEGFP-hG9a and pEGFP-ΔSET-hG9a. This work was supported by grants from the National Natural Science Foundation of China (81273545 and 81521005).

## Author contributions

Y.-f.W., M.-y.G., J.D. and Y.C. designed the experiments. Y.-f.W., J.Z., Y.S., Y.-y.S., D.-x.J. and Y.-y.H. performed the research. Y.-f.W. and Y.C. analyzed the data and wrote the paper, which was revised by M.-y.G., J.D. and Y.C. All authors approved the final version of the manuscript.
