## [Peer Review File · Nature Communications]

Reviewers' comments:

Reviewer #1 (Remarks to the Author):

On the whole a comprehensive body of work that provides a convincing body of evidence that the G9a-Hephaestin axis is important in breast tumorigenesis.

My comments are fairly minor

1. In fig 1A MDA-231 the colony number is not significantly suppressed but yet according to the blot there is no G9a. This is at odds with the right hand bar. Worth repeating in my opinion. Also the figure needs to be better described. The graphs at the bottom of fig1A and 1B what are they? If they are growth assays then it shows that shG9a1 is similarly not retarded and thus the statement in the results (stably suppressed G9a expression grow more slowly) is now not accurate.

2. Fig 2D its odd that skbr3 has both high expression of heph and g9a. Is it worth exploring deeper as to why. Also the correlation on the cell line data is not appropriate as its clear that two of the cell lines have been removed from the analyses. On the graph there are only 9 data points yet there were 11 lines in the blots. I suspect one of the lines removed was SKBR. Unless there is a good statistical reason I would be including the data for all lines into the correlation.

3. Some more detail with regards the IHC studies and how the quantitation was performed. (Fig 2E)

4. In fig 3E concerning the over expression studies only the MCF-7 cell line was shown whilst in the rest of the figure MDAs are also shown. For completeness also show the MDA data.

5. My only other comment is how the manuscript reads. Currently its quite disjointed with data provided in both supplementary and within the manuscript itself. It would be good if the most relevant supplementary could be incorporated into the body of the manuscript.

Reviewer #2 (Remarks to the Author):

The manuscript proposed an interesting new mechanism by which G9a methyltransferase regulates the breast cancer proliferation. In general, the study was well performed and in most of the cases with appropriate controls. The functional link in mechanistic claims is well documented and supported by rescue experiments which render the conclusion convincing. Furthermore, they demonstrated that G9a complexes with HDAC1 and YY1 to repress the HEPH expression which seems to have a role to play in regulating breast cancer proliferation. Finally, the authors demonstrated the relevance of the G9a-HEPH in disease progression using public data which indicates a potential prognostic role of G9a and HEPH in breast cancer.

The findings are novel though G9a's epigenetic activity in suppressing gene expression to promote cancer is expected.

Nevertheless, there are some concerns that need to be addressed to consolidate the conclusions

Major:

It is unclear why the authors chose MCF-7, MDA-MB231, and ZR-75 for knockdown or inhibitor treatment studies.

Figure 2D clearly shows that G9a levels in the above lines are much lower compared to others such as SKBR3 and MB435 cells. If the hypothesis is that G9a overexpression promotes the HEPH for proliferation advantage, shouldn't the authors use the G9a high expression lines for knockdown or inhibitor treatment experiments?

Same for the G9a inhibitors, did the inhibitors show different responses in cell lines showing different levels of G9a and HEPH ? Clearly, this is not the case as revealed in Figure 1C.

I would suggest that authors provide data on the G9a high expression lines to determine if G9a high expression lines are more susceptible to G9a inhibition or knockdown.

Minor

Figure 1, G9a knockdown does not seem to be robust in affecting cell proliferation, though the effect on colony growth appears to be obvious.

Figure 1E, in vivo data, the knockdown efficiency and the tumor growth inhibition is not well correlated.

We appreciate the reviewers' constructive suggestions to our manuscript, which greatly helped to enhance the quality of our manuscript. Our answers to the reviewer specific comments (repeated at the beginning of each paragraph and noted in bold) are:

REVIEWER 1:

1. In Fig 1A MDA-231 the colony number is not significantly suppressed but yet according to the blot there is no G9a. This is at odds with the right hand bar. Worth repeating in my opinion. Also the figure needs to be better described. The graphs at the bottom of fig1A and 1B what are they? If they are growth assays then it shows that shG9a1 is similarly not retarded and thus the statement in the results (stably suppressed G9a expression grow more slowly) is now not accurate.

Answer: We repeated the colony formation assay of MDA-MB-231 according to reviewer's suggestion. The graphs at the bottom of Fig1A and 1B are the growth curve of the indicated cells. We revised the legend of Figure 1, which will promote the clarity of the results. Since epigenetic modification is relatively slowly to have effect on cell proliferation, we prolonged the culture time in the growth assay from four days to six days, and then significant difference was exhibited. As shown in Fig 1A in the revised version, we can find that knockdown G9a in MDA-MB-231 surely suppresses cell growth and colony formation.

2. Fig 2D its odd that skbr3 has both high expression of heph and g9a. Is it worth exploring deeper as to why. Also the correlation on the cell line data is not appropriate as its clear that two of the cell lines have been removed from the analyses. On the graph there are only 9 data points yet there were 11 lines in the blots. I suspect one of the lines removed was SKBR. Unless there is a good statistical reason I would be including the data for all lines into the correlation.

Answer: We are sorry for making such a mistake. In the revised version, in order to show a more wide-ranging phenomenon, up to 20 breast cancer cell lines and human normal mammary epithelial cell MCF10A were analyzed and calculated the correlation

between G9a and HEPH. As shown in Figure 2D, G9a level was varied among different breast cancer cell lines, and really was negatively correlated with HEPH expression with P value<0.05. Additionally, we have demonstrated that depletion of G9a also stimulated significant increases in HEPH protein levels in both SKBR-3 and MDA-MB-435 cells, which with high G9a expression. The data was showed in Supplement Figure S1A, S1B.

3. Some more detail with regards the IHC studies and how the quantitation was performed.

Answer: In the IHC studies, staining of HEPH and G9a was performed as previously described with dilutions of 1:30 in HEPH and 1:100 in G9a. Two pathologists blinded to patient's outcome, supporting clinical data and prospectives reviewed these cases and scored the slides by Imagescope using the same scale, with 0 representing low or undetectable staining, 1 representing intermediate staining, and 2 representing intense staining. Signal localization (cytoplasmic, nuclear) and the staining intensity was quantified using H-score (histochemical score) analysis considering the tumour component only. Each antigen staining intensity and positive rate was scored independently in three tumor clusters each containing at least 20 cancer cells. The chi-square test was done on dyeing rate and dyeing composite index (mean positive staining intensity*dyeing rate) respectively. The survival analysis is based on the median cutoff value of dyeing composite index and adopts Kaplan Meier survival analysis method and the log-rank statistical test for lifetime single factor analysis.

4. In Fig 3E concerning the over expression studies only the MCF-7 cell line was shown whilst in the rest of the figure MDAs are also shown. For completeness also show the MDA data.

Answer: The data of HEPH over-expression in MDA-MB-231 were added in Figure 3F in the revised version.

5. My only other comment is how the manuscript reads. Currently its quite

disjointed with data provided in both supplementary and within the manuscript itself. It would be good if the most relevant supplementary could be incorporated into the body of the manuscript.

Answer: We reorganized the figures, and tried our best to put the most relevant data into the body of the manuscript. Such as “the data of HEPH interference or over-expression” now is in the main body. Other minor adjustments are presented in the revised paper to make the article more coherent and accessible.

REVIEWER 2:

1. It is unclear why the authors chose MCF-7, MDA-MB231, and ZR-75 for knockdown or inhibitor treatment studies

Answer: Such a decision was made based on our integrated results. At first, it is according to our breast cancer cell lines profiling (Figure 2D). Compared with normal breast epithelial cell MCF10A, we found that all tested breast cancer cell lines, such as MCF-7, MDA-MB-231 and ZR-75-30 cells have higher level of G9a. In addition, these three cell lines represent three different human breast cancer subtypes (ER positive, triple negative, and Her2 positive) respectively. And they are all commonly used in breast cancer research. So the similar data from these three different cell lines should indicate that HEPH expression repressed by G9a is independent of breast tumor type.

2. Figure 2D clearly shows that G9a levels in the above lines are much lower compared to others such as SKBR3 and MB435 cells. If the hypothesis is that G9a overexpression promotes the HEPH for proliferation advantage, shouldn't the authors use the G9a high expression lines for knockdown or inhibitor treatment experiments?

Answer: It is really an important question. We did notice that SKBR-3 and MDA-MB-435 cells have high level of G9a in our cell profiling assay. We had generated stable cell lines with limited expression in SKBR-3 and transiently silenced

G9a in MDA-MB-435 cells. Then we performed a series of experiments to examine G9a's effect on cell growth and HEPH expression in these two cell lines. No doubt, similar data were displayed. Knockdown G9a in SKBR-3 and MDA-MB-435 also elevates HEPH expression level and has a tremendous effect on cell growth. These data were displayed in the supplementary data Figure. S1 in this version.

3. Same for the G9a inhibitors, did the inhibitors show different responses in cell lines showing different levels of G9a and HEPH ? Clearly, this is not the case as revealed in Figure 1C. I would suggest that authors provide data on the G9a high expression lines to determine if G9a high expression lines are more susceptible to G9a inhibition or knockdown.

Answer: As suggested, the effects of G9a inhibitors on HEPH expression and cells proliferation were confirmed in ten more breast cancer cell lines (Figure 1C, S2B). We found that the specific inhibitors really inhibited cells growth and increased HEPH levels in these cells, but the responses are not well associated with G9a levels. The inconsonant effect may be due to G9a non-histone methyltransferase activity. However, G9a high expression lines, such as SKBR3 and MDA-MB-435, are more susceptible to G9a knockdown(Figure S1A, S1B). The detailed mechanisms deserve further elucidation.

4. Figure 1, G9a knockdown does not seem to be robust in affecting cell proliferation, though the effect on colony growth appears to be obvious.

Answer: The colony growth inhibition really appeared more obviously in our study. We thought it is due to time. In the colony formation assay, cells were seeded onto 6-well plates, and had cultured for 10-15 days until the colonies became visible. But in the proliferation assay, cells had been cultured only for 96 hours. Our new data showed that if we prolong the culture time to six days, the effect of G9a knockdown on cell proliferation also appears to be obvious.

5. Figure 1E, *in vivo* data, the knockdown efficiency and the tumor growth inhibition is not well correlated.

Answer: In the *in vivo* tumorigenic assay (Figure 1E, F), there were 6 mice in each

group. In this version, we analyzed all the six samples, and found that not only the G9a interference efficiency in S1 shG9a1 was slightly better than shG9a3, but also the reduction of H3K9me2 and the elevation of HEPH level is remarkably evident than shG9a3 (Figure 1F). What's more, according to our previous data, the iron content in tumor tissue from S1 shG9a1 xenografts was much lower than shG9a3 xenografts (Figure S3B). Thus we thought that the knockdown efficiency and the tumor growth inhibition is correlated.

REVIEWERS' COMMENTS:

Reviewer #1 commented for the editors only and was satisfied by the revision.

Reviewer #2 (Remarks to the Author):

the authors have addressed most of the comments but it is wise to include the some of the additional key data in the main manuscript rather than supplemental

We appreciate the reviewers' suggestions to our manuscript. Our answers to the reviewer specific comments (repeated at the beginning of each paragraph and noted in bold) are:

REVIEWER 1: commented for the editors only and was satisfied by the revision.

REVIEWER 2: The author have addressed most of the comments but it is wise to include the some of the additional key data in the main manuscript rather than supplemental.

Answer: We rearranged the supplemental figures according to the reviewer suggestion. In this version, the supplemental figure1 was rearranged into Figure 1.